# The Prognostic and Predictive Utility of CDX2 in Colorectal Cancer

**DOI:** 10.3390/ijms25168673

**Published:** 2024-08-08

**Authors:** Wei Yen Chan, Wei Chua, Kate Wilkinson, Chandika Epitakaduwa, Hiren Mandaliya, Joseph Descallar, Tara Laurine Roberts, Therese Maria Becker, Weng Ng, Cheok Soon Lee, Stephanie Hui-Su Lim

**Affiliations:** 1Liverpool Cancer Therapy Centre, Liverpool Hospital, Liverpool, NSW 2170, Australia; weiyen_chan@hotmail.com (W.Y.C.); wei.chua@health.nsw.gov.au (W.C.); kate.wilkinson1@health.nsw.gov.au (K.W.);; 2School of Medicine, Western Sydney University, Locked Bag 1797, Penrith, NSW 2571, Australia; tara.roberts@westernsydney.edu.au (T.L.R.); therese.becker@inghaminstitute.org.au (T.M.B.); soon.lee@westernsydney.edu.au (C.S.L.); 3Ingham Institute for Applied Medical Research, Liverpool, NSW 2170, Australia; joseph.descallar@health.nsw.gov.au; 4Department of Anatomical Pathology, Liverpool Hospital, Liverpool, NSW 2170, Australia; dilanchandika@gmail.com; 5Macarthur Cancer Therapy Centre, Campbelltown Hospital, Campbelltown, NSW 2560, Australia; hiren.mandaliya@calvarymater.org.au; 6Faculty of Medicine, South Western Sydney Clinical School, The University of New South Wales, Liverpool, NSW 2170, Australia

**Keywords:** colorectal cancer, caudal type homeobox transcription factor 2, biomarkers, prognostic, predictive, chemotherapy

## Abstract

**Simple Summary:**

Caudal type homeobox transcription factor 2 (CDX2) is a protein expressed in the normal intestinal epithelium that is essential for regulating cell growth. It has been found to be downregulated or lost in a significant proportion of patients with colorectal cancer and can influence cancer behaviour and treatment response. Therefore, further research and assessment of CDX2 levels in tumour tissues is crucial for identifying potential markers that can select patients with worse outcomes and guide treatment decisions to improve survival. This study found that tumours with low CDX2 expression were associated with more aggressive features of cancer and worse outcomes for patients. However, giving additional chemotherapy after surgery to certain early-stage colon cancer patients with low CDX2 levels did not significantly improve survival. These findings support CDX2 as a marker for colorectal cancer prognosis, but more research is needed to determine its ability to predict chemotherapy outcomes.

**Abstract:**

Caudal type homeobox transcription factor 2 (CDX2) is a gastrointestinal cancer biomarker that regulates epithelial development and differentiation. Absence or low levels of CDX2 have been associated with poor prognosis and proposed as a chemotherapy response predictor. Tumour tissue samples from 668 patients with stage I–IV colorectal cancer were stained for CDX2 and stratified into two subgroups according to expression levels. Statistical tests were used to evaluate CDX2’s relationship with survival and chemotherapy response. Of 646 samples successfully stained, 51 (7.9%) had low CDX2 levels, and 595 (92.1%) had high levels. Low CDX2 staining was associated with poor differentiation and the presence of lymphovascular or perineural invasion and was more common in colon and right-sided tumours. Overall survival (*p* < 0.001) and disease-free survival (*p* = 0.009) were reduced in patients with low CDX2 expression. Multivariable analysis validated CDX2 as an independent poor prognostic factor after excluding confounding variables. There was no statistically significant improvement in survival with adjuvant chemotherapy in stage II colon cancer (*p* = 0.11). In the rectal cohort, there was no relationship between CDX2 levels and therapy response. While confirming the prognostic utility of CDX2 in colorectal cancer, our study highlights that larger studies are required to confirm its utility as a predictive chemotherapy biomarker, especially in left-sided and rectal cancers.

## 1. Introduction

Colorectal cancer (CRC) is the fourth most frequently diagnosed cancer and second most common cause of cancer death in Australia [1,2]. The estimated risk of being diagnosed with colorectal cancer by the age of 85 is 5.2% [1]. The survival rate of CRC varies significantly between stages. For instance, patients with stage III CRC have a 5-year survival rate of 72%, while those with stage IV disease have a much lower survival rate of 13% [3]. The current standard of care for management of stage III colon cancer consists of surgery followed by adjuvant (post-operative) chemotherapy to eradicate micrometastases. However, the absolute magnitude of benefit from adjuvant chemotherapy in resected stage II disease is not as clear [4,5]. The decision of whether to use adjuvant chemotherapy is informed in part by the presence or absence of high-risk clinicopathologic features; therefore, identification and validation of biomarkers for CRC is essential for predicting high-risk patients who may benefit from additional chemotherapy. 

Several biomarkers are currently in use to provide a more comprehensive assessment of tumours in colorectal cancer, such as carcinoembryonic antigen (CEA), mismatch repair deficiency (MMR), microsatellite instability (MSI), v-raf murine sarcoma viral oncogene homolog B1 (BRAF), reticular activating system (RAS) mutations and caudal-type homeobox transcription factor 2 (CDX2) [6,7]. CDX2 is a transcription factor that is essential for intestinal development and has been proposed as a potential CRC biomarker [8]. The CDX2 protein family belongs to the ParaHox gene cluster and consists of three members: CDX1, CDX2 and CDX4 [9,10]. CDX2 regulates genes that control enterocyte maturation, including mucin-2 (MUC2). It is integral to intestinal cell differentiation as well as maintenance of cell integrity [11,12]. Consequently, CDX2 deficiency results in defective barrier function and enterocyte atrophy. It is thought that loss of CDX2 is a result of epigenetic mechanisms rather than genomic alterations [13]. CDX2 loss results in downregulation of p21 and thus affects crucial pathways in CRC carcinogenesis, including MAPK, Wnt and TGF-beta [14].

Immunohistochemical (IHC) detection of CDX2 expression is clinically useful as a relatively specific marker for intestinal epithelial differentiation [15]. Strong, diffuse CDX2 staining is typically observed in the normal intestinal epithelium, including that of the duodenum, ileum, appendix, colon and rectum, but can also be found in malignant tissue. The highest frequency of extensive CDX2 staining is found in colorectal adenocarcinomas (90%), followed by ampullary (33%) and gastroesophageal adenocarcinomas (20%) [15]. Therefore, changes in CDX2 expression are a valuable tool for understanding disease progression and prognosis. CDX2 has been found to be downregulated or lost in a proportion of CRC [16]. Lack of CDX2 expression is associated with more aggressive features such as advanced stage, poor differentiation, vascular invasion and BRAF mutation, which can contribute to poorer survival [17,18,19]. Therefore, evaluation of CDX2 expression can provide important prognostic information to allow risk stratification and personalisation of therapy. Moreover, it is a readily available and economical test that can be readily applied to clinical practice.

Ideally, biomarkers that can be easily measured on surgical specimens would be beneficial in providing information on the biological characteristics of tumours beyond the clinical TNM classification system. This is because TNM staging assumes that all tumours within the same stage behave similarly and does not account for differences in tumour biology. There is some evidence to suggest that (i) patients with CDX2-negative tumours have lower survival rates than those with CDX2-positive tumours and that (ii) the loss of CDX2 expression can identify a subset of patients with stage II colon cancer who are at high risk for disease recurrence and may benefit from adjuvant chemotherapy [20]. These findings highlight the potential clinical utility of CDX2 as a prognostic (providing information about likelihood of disease progression or outcome regardless of treatment) and predictive (providing information about likelihood of response to a particular therapy) biomarker in CRC and emphasises the importance of incorporating biomarker testing in routine practice [21]. 

In this retrospective study, we assessed a cohort of patients with stage I to IV colorectal cancer for CDX2 staining and correlated it with patient demographics; clinicopathologic features; and cancer outcomes including overall survival (OS), disease-free survival (DFS) and recurrence-free survival (RFS). Most studies to date have evaluated colon cancer alone or colorectal cancer as a combined entity, with minimal focus on the rectal cohort specifically. This study investigated CDX2’s prognostic and predictive value, its potential role in clinical practice and whether this differs between colonic and rectal tumours.

## 2. Results

### 2.1. Study Population 

A total of 668 patients were included in this study. A description of patient characteristics is detailed in Table 1. The cohort was well balanced between males (58.7%) and females (41.3%). Mean age at diagnosis was 68 years (ranging from 23 to 96 years). Almost half the study population was aged above 70 years (49.1%). regarding ethnicity, Caucasians were the majority at 86.4%, followed by Asians at 11.4%. Notably, our study cohort reflects a significantly culturally and linguistically diverse population, particularly due to its substantial proportion of Asian individuals [22].

Although all cancer stages were included, stage II and stage III were the most prominent cohort in the study, contributing 31% and 37.4%, respectively. Adenocarcinoma was the predominant histological subtype (98.4%). With regard to tumour location, 60.8% of patient tumours originated in the colon, and 39.2% originated in the rectum. A total of 595 patients (92.1%) in the studied cohort had high CDX2 expression, whereas only 51 patients (7.9%) had low CDX2 expression. A total of 22 patients (3.3%) had no tissue or missing data on CDX2 expression; hence, a total of 646 patients were analysed for detected CDX2 expression.

A lower proportion of stage II colon cancer patients received adjuvant chemotherapy (at 17.3%) than stage III colon cancer patients (at 63%: 26% single-agent chemotherapy, 37% doublet). Among the stage III colon cancer cohort, notably, those who received adjuvant chemotherapy were younger (mean age 63 years) than those who did not (mean age 77). Importantly, there was no significant difference observed in key pathological features including CDX2 expression, pathological subtype, tumour differentiation and invasiveness between the two groups. Of the rectal cohort, most proceeded to upfront surgery (76.3%), with only 23.7% having any neoadjuvant therapy, including short-course radiotherapy and long-course chemoradiotherapy.

### 2.2. Association of CDX2 Staining with Clinicopathological Features

The patient subgroups were examined to determine whether there was an association with CDX2 staining and the clinicopathological features of the study population. Table 2 presents descriptive statistics by CDX2 expression for 595 patients with CDX2-high tumours and 51 patients with CDX2-low tumours. Two clinicopathological features that were significantly associated with CDX2 expression were tumour differentiation (*p* < 0.001) and presence of lymphovascular or perineural invasion (*p* = 0.002). In terms of sidedness, the rate of low CDX2 expression was numerically higher in the right-sided tumours (11.7%) than the left-sided tumours (6.0%). Similarly, there was a higher number of tumours with low CDX2 expression in the colon (9.2%) than in the rectum (5.7%); however, this relationship with CDX2 was not statistically significant. CDX2 expression did not differ by ethnicity. 

### 2.3. Low CDX2 Expression Correlates with Poorer Prognosis

Low CDX2 expression was associated with poorer OS (*p* = 0.002), as shown in Figure 1. This was consistent across all stages (*p* < 0.001). Low-CDX2-expression tumours were also associated with lower rates of disease-free survival (*p* = 0.009), cancer-specific survival (*p* < 0.001) and recurrence-free survival (*p* = 0.004). There was no statistically significant difference in overall survival (*p* = 0.522) or disease-free survival (*p* = 0.126) based on tumour location, i.e., colon or rectum (Figure 2). In a multivariable analysis that adjusted for tumour stage, sex and location as confounding variables, the association between low-CDX2-expression colorectal tumours and a lower rate of overall survival remained significant (Table 3). 

### 2.4. CDX2 as a Predictive Biomarker for Adjuvant Chemotherapy Benefit in Stage II and III Colon Cancer

In addition to investigating CDX2 expression as a prognostic biomarker, we examined its relationship with adjuvant chemotherapy to determine whether we could predict patients who might benefit from adjuvant chemotherapy (Figure 3). Among all patients with stage II colon cancer (*p* = 0.339) and the high-CDX2 subgroup (*p* = 0.656), adjuvant chemotherapy was not associated with improvement in overall survival. In the low-CDX2 subgroup, there was a trend towards improved overall survival with adjuvant chemotherapy compared to no chemotherapy; however, this was not statistically significant (*p* = 0.113). A test for interaction between CDX2 level and adjuvant chemotherapy in stage II colon cancer was not statistically significant (*p* = 0.98). Among patients with stage III colon cancer, there was a statistically significant improvement in the rate of overall survival for the entire cohort (*p* < 0.001) even after adjusting for disease stage, sex and sidedness. This improvement was evident in both the low-CDX2 subgroup (HR 0.05, [CI 0.01–0.23], *p* < 0.001) and the high-CDX2 subgroup (HR 0.19 [CI 0.11–0.32]), *p* < 0.0001) when adjuvant chemotherapy was administered.

### 2.5. CDX2 as a Predictive Biomarker in Rectal Cancer

In the rectal cohort, neoadjuvant therapy consisted of short-course radiotherapy and long-course chemoradiotherapy. There was no statistically significant association between the level of CDX2 expression and overall survival (*p* = 0.825) in rectal patients who received neoadjuvant therapy. There was an improvement in overall survival in rectal cancer patients who were treated with adjuvant chemotherapy regardless of high (*p* = 0.017) or low CDX2 expression (*p* < 0.0001). 

## 3. Discussion

Precise risk stratification based on reliable prognostic markers is critical for guiding treatment decisions and accurately predicting outcomes in patients with CRC. In this study, we identified the pattern of CDX2 expression in stage I–IV CRC and correlated this with demographics, clinicopathologic features, response to adjuvant chemotherapy and survival outcomes. In addition, we examined these parameters in colon and rectal cancer patients separately. We showed that low CDX2 expression was significantly correlated with poor prognostic features including lack of tumour differentiation, presence of lymphovascular or perineural invasion and poorer OS and DFS outcomes in both colon and rectal cancers. These findings are consistent with the literature and support CDX2 expression status as a potentially valuable prognostic marker for identifying patients with more aggressive disease and poorer clinical outcomes [23]. 

In addition to evaluating the prognostic value of CDX2 expression, we also investigated its potential predictive role. In stage II colon cancer, the benefit of adjuvant chemotherapy is small at approximately 5% [24,25]; hence, having a predictive biomarker may assist clinicians in determining which patients are most likely to benefit. In stage II colon cancer, we observed a trend towards improved survival with addition of adjuvant chemotherapy in low-CDX2 patients; however, this association did not reach statistical significance. Notably, our findings differ from those of a previous study by Dalerba et al., who reported a significant benefit of adjuvant chemotherapy in this cohort [20]. However, we acknowledge that the number of patients with stage II colon cancer and low CDX2 expression who had adjuvant chemotherapy in our study was small (fewer than 10 patients), compared to the study by Dalerba et al., where there were 25 patients. Although this study was not powered to demonstrate predictive value due to small numbers, it was indicative of a trend that tumours with low CDX2 expression may select stage II patients who are more likely to benefit from adjuvant chemotherapy. Nevertheless, larger studies are needed to confirm CDX2’s predictive utility. 

In stage III colon cancer, there was a statistically significant overall survival benefit with adjuvant chemotherapy in both high- and low-CDX2-expression tumours. A study by Bruun et al. suggested that in stage III patients, the 5-year RFS was higher amongst patients with loss of CDX2 expression who received adjuvant chemotherapy (61%) than those who did not (44%) which supports our results [26]. However, this is less clinically meaningful, as most stage III colon cancer patients would be recommended for adjuvant chemotherapy regardless of CDX2 expression. 

This study also focused on risk prediction with CDX2 in rectal cancer, as there is less available literature on this, with most grouping colon and rectal cancers into one cohort. The lack of significant differences in survival outcomes between patients with high and low CDX2 expression who received adjuvant chemotherapy for rectal cancer suggests that CDX2 expression likely does not predict response to chemotherapy in this population. Similarly, there was no correlation between CDX2 expression and treatment outcomes in the rectal cancer cohort that received neoadjuvant therapy. This is an important finding, as CDX2 cannot be recommended as a reliable predictor of neoadjuvant treatment and adjuvant chemotherapy efficacy in rectal cancer. Adjuvant chemotherapy is an effective treatment option for rectal cancer patients, regardless of CDX2 expression status. Based on our findings, CDX2 is not a useful predictive biomarker for adjuvant chemotherapy in rectal cancer. In addition, it is unclear to what extent adjuvant therapy contributes to the observed outcomes, given that some patients had had treatment pre-operatively. Nevertheless, the interpretation of these findings in the current clinical context is challenging due to significant changes in the treatment landscape of rectal cancer over the last few years, with a shift towards total neoadjuvant therapy [27]. 

Heterogeneity within colorectal cancer has resulted in variable patient responses to classic systemic therapies. One approach to defining the heterogeneity is to use a framework such as the consensus molecular subtypes (CMS) to provide a systematic way of classifying colorectal cancer based on genetic and molecular characteristics [28]. Through more nuanced insight into tumour biology, we can better predict responses to therapeutic alternatives to select the most appropriate therapy. CDX2 has been linked to the CMS system; the lack of CDX2 expression can assist in differentiating between subtypes, as it is only present in the mesenchymal subgroup (CMS4) [29]. CDX2 is thought to suppress the epithelial–mesenchymal transition (EMT); hence, the loss of CDX2 results in EMT [30]. Thus far, the implementation of CMS in trials and clinical practice has been limited by the requirement of comprehensive genomic analysis. To overcome this limitation, IHC detection can be used as a surrogate to enable more practical integration of CMS into clinical research and patient care [28,31]. 

One limitation of our retrospective study pertains to its data source, specifically its dependence on a surgical database. Consequently, our study exclusively included patients who underwent surgical interventions and had corresponding pathology slides available, thereby excluding those who did not undergo surgical procedures. Additionally, it is essential to acknowledge the challenge posed by the lack of standardisation of CDX2 staining, which makes data interpretation between studies and the clinical use of this biomarker in colorectal cancer inherently complex. There is significant variability in methods used for staining, including the type of antibody, the staining protocol and the interpretation of staining results. Different antibodies (e.g., clone CDX2–88 BioGenex, CellMarque, DAKO M3636, A1629 ABclonal) may have different binding specificities and sensitivities, which can affect the accuracy of the staining result [32,33]. The scoring system used to interpret CDX2 staining results can also vary between laboratories and pathologists, which can lead to differences in assignment of CDX2 expression. For our study, staining interpretation of CDX2 was given a score based on percentage of deeply stained nuclei and then classified as high or low, whereas other studies used positive or negative; normal or absent; or none, weak, moderate or strong [34]. We opted for this scoring to account for active CDX2, as nuclear localisation is a prerequisite for transcription factor activity. To address scoring variations, the International Collaboration on Cancer Reporting (ICCR) has developed guidelines for standardised reporting of colorectal cancer biomarkers; however, CDX2 is yet to be included [35]. A recent expert review has also elucidated the limitations of CDX2 staining [9].

The inconsistencies in methodologies across studies have also led to a wide range of reported loss of CDX2 expression. Our results are comparable, with 7.6% showing CDX2 loss. This is consistent with literature where the rate of CDX2 loss/absence/low ranged from 5–30% [36]. In our study, there was an increase in the percentage of cases with loss of CDX2 expression as the cancer stage advanced, i.e., stages I (3.5%), II (5.0%), III (10.7%) and IV (12.1%), which was statistically significant (*p* = 0.015). The existing literature shows a similar pattern of expression, with loss of CDX2 associated with advanced stage [37]. Some studies also suggest that CDX2 expression is higher in left-sided than right-sided colorectal cancer, which is concordant with our results showing that high CDX2 expression was higher in the left-sided (94%) than right-sided samples (88%) [38]. The reason for this is unclear but may be related to differences in the embryonic origin and molecular characteristics of the left and right colon. In our study, low CDX2 expression was observed more commonly in right-sided (11.7%) than left-sided tumours (6.0%), which was statistically significant. More research is needed to fully understand the implications of this difference for clinical practice; nonetheless, it is established that right-sided colon cancers generally have a worse prognosis. As such, loss of CDX2 expression may simply be linked to unfavourable prognostic features. 

## 4. Materials and Methods

### 4.1. Patients and Surgical Specimens

A database of surgical specimens collected from patients with stage I to IV colorectal cancer during the period 2000 to 2011 was retrospectively analysed and stained for CDX2. The search yielded 668 specimens, including 406 of colonic origin (from 2006 to 2011) and 262 of rectal origin (from 2000 to 2011). Right-sided tumours (32.5%) were considered to include the proximal colon up to the splenic flexure, and left-sided tumours (67.5%) were defined as being located from the splenic flexure to the rectum. Biological characteristics of patient tumours was correlated with pathological stage, patient demographics and treatment details. This study was conducted with approval of the South Western Sydney Local Health District Human Research Ethics Committee (HREC Reference: HREC/12/LPOOL/102). 

### 4.2. Tissue Microarrays and Immunohistochemical Analysis

In this retrospective analysis, tissue microarray (TMA) slides from archival tissue samples of patients who underwent surgical resection of their primary colorectal cancer were assessed. Tissue samples were mapped according to the following locations: normal tissue close to the tumour, normal tissue away from the tumour, tumour centre (average score of 2 cores), tumour periphery (average score of 2 cores), lymph node metastasis (average score of 2 cores), polyp and adenoma. The tumour periphery was selected as the primary marker for analysis. The choice of the tumour periphery was due to its biological relevance as an area of active invasion and interaction with the microenvironment, reflecting tumour aggressiveness and potential clinical behaviour. 

CDX2 protein expression was evaluated using formalin-fixed paraffin-embedded (FFPE) tissue sections, which were stained with a validated assay of mouse CDX2 monoclonal antibody (clone CDX2–88, BioGenex) at a concentration of 4 mg per millilitre using an automated staining platform. All tissue microarrays were scored for CDX2 expression by two independent pathologists who were blinded to the patients’ details. The scorers underwent a period of training with a multiheaded microscope to ensure consistent and reliable interpretation. With a test series of at least 36 tissue core sections, intra-observer and interobserver agreement were estimated by use of kappa (j) and Spearman’s rho (q). Training was ended when the desired level of agreement, consistent over time, was achieved (j > 0.6 and q > 0.8). 

An average score was obtained from the duplicate cores of each tissue sample. A score of 1 to 5 was assigned according to the proportion of deeply stained nuclei (Table 4). Only deep nuclear staining and not cytoplasmic staining was considered for scoring. This selection was made with the understanding that the active-state protein, being a transcription factor, primarily resides in the nucleus [36]. Tissue cores from areas of the same tumour were paired at the end, and if the scores were discordant, the final score for the tumour was upgraded to the higher score. Patients were stratified into two subgroups based on CDX2 expression: high (for a score of 5, i.e., 75–100% deeply stained nuclei) and low (for scores 1 to 4, i.e., less than 75% deeply stained nuclei) (Figure 4). Staining of normal tissue adjacent to and distant from the tumour was also conducted for our internal control group. Our findings revealed that over 95% of normal colonic and rectal tissues exhibited high CDX2 expression, as expected. 

### 4.3. Statistical Analysis

Statistical analysis was conducted using SAS Enterprise Guide version 8.2 and R statistical software version 4.2.2. Descriptive statistics were used to characterise the patient population. To evaluate the association between CDX2 staining and survival outcomes, the Kaplan–Meier method and the log-rank test were used to compare the two subgroups. Overall survival (OS) was defined as time from diagnosis to death from any cause or date of last follow-up, disease-free survival (DFS) as time from diagnosis until first evidence of disease progression or death, recurrence-free survival (RFS) as time from diagnosis to recurrence and cancer-specific survival (CSS) as time from diagnosis to death from disease [39]. 

Univariate analysis was performed using a one-sample *t*-test, and *p* < 0.05 was considered statistically significant. Multivariable Cox regression was performed to determine whether the association between low-CDX2 cancers and poorer overall survival remained significant after adjusting for confounding variables. The covariates included were pathological tumour stage, sex, tumour side, site of disease and CDX2 expression. In the subset of patients with stage II and III disease, we used a multivariable model. This analysis included confounding variables from the previous model and added adjuvant chemotherapy as a covariate. Additionally, we explored interactions between CDX2 expression level, disease stage, tumour location (colon or rectum) and adjuvant chemotherapy to determine the impact of adjuvant therapy on overall survival. 

## 5. Conclusions

Overall, our study findings align with the existing literature, providing further evidence for the correlation between low CDX2 expression and poorer overall survival in both colon and rectal cancers. However, we cannot draw definitive conclusions regarding the predictive value of CDX2 as a biomarker for colorectal cancer, especially in rectal or left-sided tumours. Low CDX2 expression is associated with aggressive features such as lack of tumour differentiation and presence of lymphovascular invasion, and it has an increased frequency in advanced-stage and right-sided tumours. The combination of promising data and a lack of standardisation in CDX2 staining and interpretation supports further efforts to standardise international reporting; such standards need to be in place before this promising biomarker can be used routinely in clinical practice. Molecular analyses examining the potential epigenetic changes associated with CDX2 loss may complement the utility of CDX2 as an immunohistochemical biomarker.

## Figures and Tables

**Figure 1 ijms-25-08673-f001:**
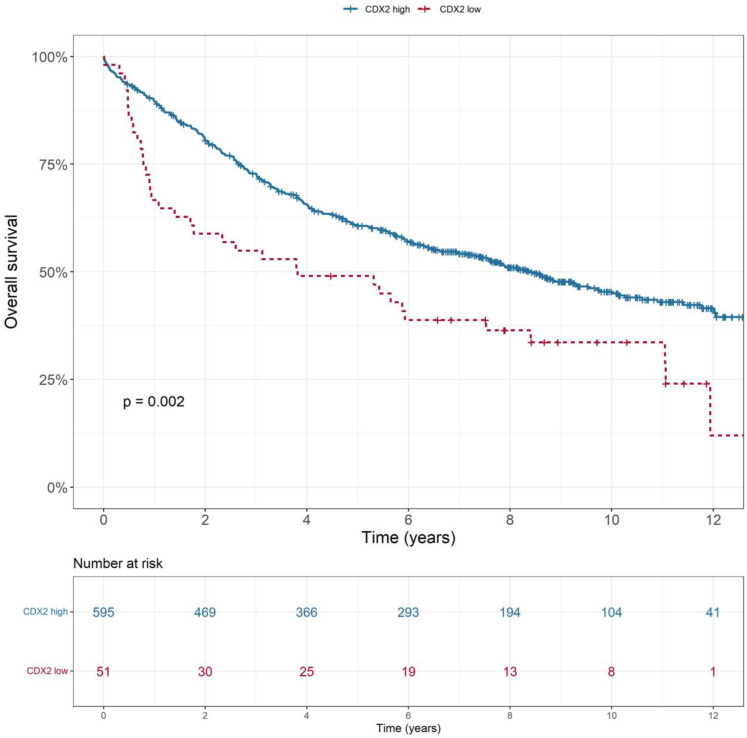
Overall survival across all stages of colorectal cancer, stratified by level of CDX2 expression.

**Figure 2 ijms-25-08673-f002:**
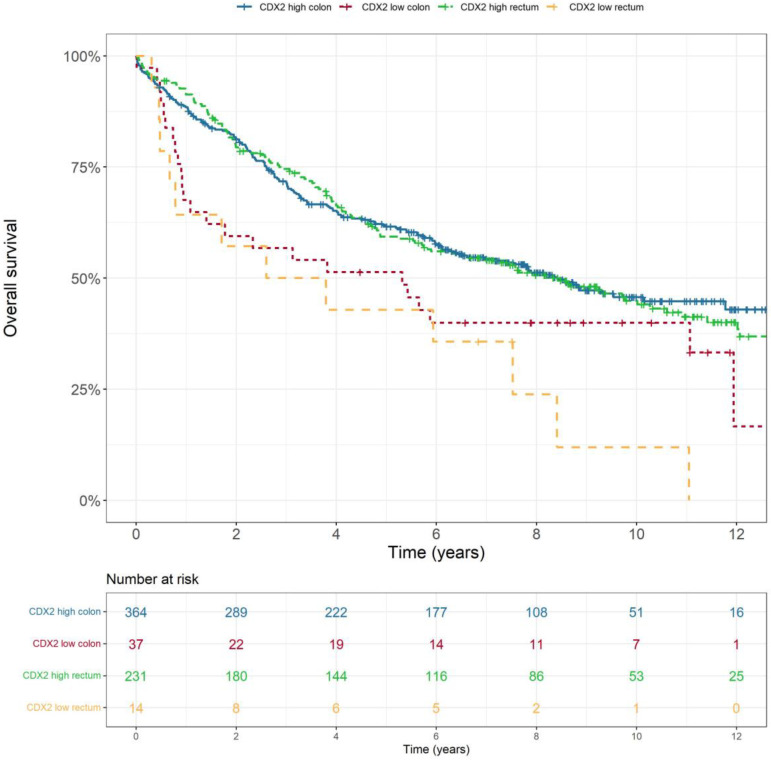
Overall survival according to tumour location (colon and rectum), stratified by CDX2 expression (high and low).

**Figure 3 ijms-25-08673-f003:**
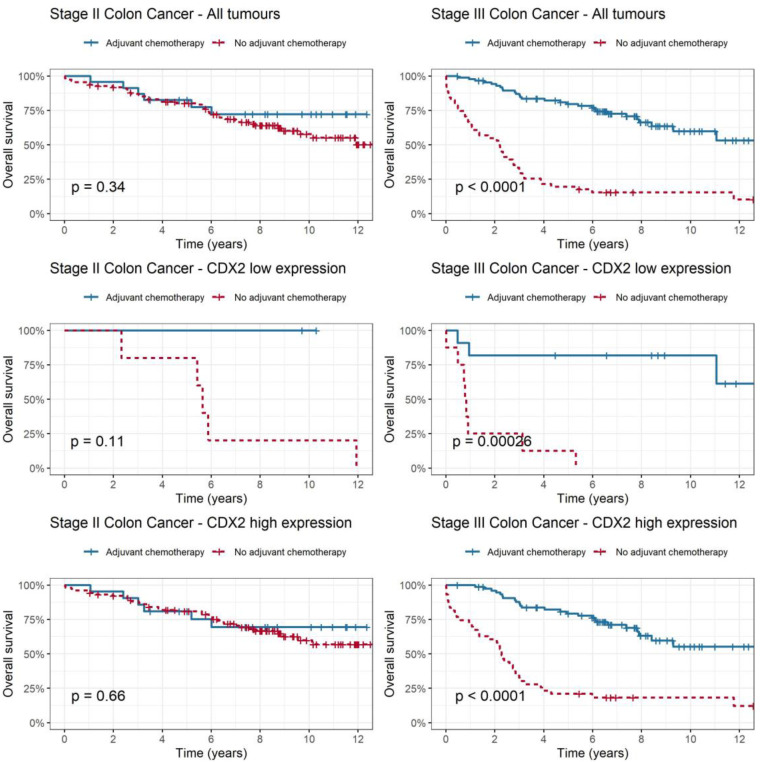
Relationship between CDX2 expression and benefit from adjuvant chemotherapy. Statistical significance was determined using the Kaplan–Meier method and the log-rank test to generate *p*-values.

**Figure 4 ijms-25-08673-f004:**
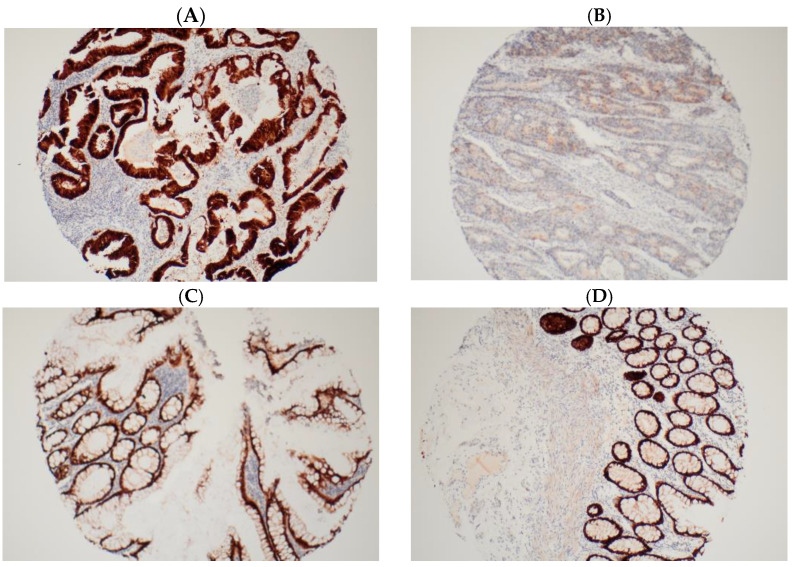
Immunohistochemical staining images of CDX2 on tumour tissue microarray. (**A**) High CDX2 (score 5) staining in tumour periphery. (**B**) Low CDX2 (score 1) staining in tumour periphery. (**C**) Normal tissue close to tumour with high CDX2 (score 5) staining. (**D**) Normal tissue away from tumour with high CDX2 (score 5) staining. Magnification ×100.

**Table 1 ijms-25-08673-t001:** Patient characteristics of study population.

Variable	Class	Total (*n* = 668)			Percentage (%)
Sex	Male	392			58.7
	Female	276			41.3
CDX2 expression	High (5)	595			89.1
	Low (1–4)	51			7.6
	Missing	22			3.3
Ethnicity	Caucasian	577			86.4
	Asian	76			11.4
	Pacific Islander	8			1.2
	African	3			0.4
	Aboriginal/Torres Strait Islander	4			0.6
Mean age at diagnosis	68.6 (23–96)				
Age at diagnosis	≤70 years	340			50.9
	>70 years	328			49.1
Cancer stage		Colon	Rectum	Total	
	I	77	39	116	17.4
	II	133	74	207	31.0
	III	138	112	250	37.4
	IV	58	37	95	14.2
Pathology	Adenocarcinoma	657			98.4
	Mucinous	9			1.3
	Signet ring	2			0.3
Cancer side	Right	217			32.5
	Left	451			67.5
Colon vs. rectum	Colon	406			60.8
	Rectum	262			39.2
Tumour differentiation	Poor	98			14.7
	Good/moderate	570			85.3
Lymphovascular or perineural invasion	Absent	438			65.8
	Present	228			34.2
Colon					
Stage II adjuvant chemotherapy	No	109			82.6
	Yes	23			17.4
Stage III adjuvant chemotherapy	None	51			37.5
	Single agent	36			26.5
	Doublet	49			36.0
Stage IV palliative chemotherapy	No	20			34.5
	Yes	38			65.5
Rectum					
Neoadjuvant therapy	No	200			76.3
	Yes	62			23.7
Adjuvant chemotherapy	No	123			54.2
	Yes	104			45.8

**Table 2 ijms-25-08673-t002:** Patient characteristics based on CDX2 expression.

Variable	Class	CDX2 Expression		
		High, *n* = 595 (%)	Low, *n* = 51 (%)	*p*-Value
Sex	Male	356 (94.4)	21 (5.6)	0.012
	Female	239 (88.8)	30 (11.2)	
Ethnicity	Caucasian	511 (91.9)	45 (8.1)	0.373
	Asian	71 (94.7)	4 (5.3)	
	Pacific Islander	7 (87.5)	1 (12.5)	
	African	3 (100.0)	0 (0.0)	
	Aboriginal/Torres Strait Islander	3 (75.0)	1 (25.1)	
Age at diagnosis	≤70 years	288 (92.3)	24 (7.7)	0.885
	>70 years	307 (91.9)	27 (8.1)	
Cancer stage	I	110 (96.5)	4 (3.5)	0.015
	II	189 (95.0)	10 (5.0)	
	III	216 (89.3)	26 (10.7)	
	IV	80 (87.9)	11 (12.1)	
Pathology	Adenocarcinoma	587 (92.2)	50 (7.8)	0.525
	Mucinous	7 (87.5)	1 (12.5)	
	Signet ring	1 (100.0)	0 (0.0)	
Cancer side	Right	189 (88.3)	25 (11.7)	0.019
	Left	406 (94.0)	26 (6.0)	
Colon vs. rectum	Colon	364 (90.8)	37 (9.2)	0.132
	Rectum	231 (94.3)	14 (5.7)	
Tumour differentiation	Poor	74 (77.1)	22 (22.9)	<0.001
	Good / moderate	521 (94.7)	29 (5.3)	
Lymphovascular or perineural invasion	Absent	399 (94.5)	23 (5.5)	0.002
	Present	194 (87.4)	28 (12.6)	
Colon stage II	Received adjuvant chemotherapy	21 (91.3)	2 (8.7)	0.607
	Did not receive adjuvant chemotherapy	102 (95.3)	5 (4.7)	
Colon stage III	Received adjuvant chemotherapy	76 (87.4)	11 (12.6)	0.618
	Did not receive adjuvant chemotherapy	43 (84.3)	8 (15.7)	

**Table 3 ijms-25-08673-t003:** Multivariable analysis to evaluate the relationship between CDX2 level and overall survival whilst adjusting for confounding variables including tumour stage, sex and tumour location.

Subgroup	Univariate Analysis		Multivariable Analysis	
	Hazard Ratio (95% CI)	*p*-Value	Hazard Ratio (95% CI)	*p*-Value
All patients (*n* = 668)				
CDX2-negative	0.584 (0.413–0.825)	0.002	0.602 (0.421–0.86)	0.005
Tumour stage				
Stage I vs. stage IV	0.117 (0.080–0.172)	<0.001	0.118 (0.08–0.173)	<0.001
Stage II vs. stage IV	0.125 (0.091–0.172)	<0.001	0.123 (0.089–0.17)	<0.001
Stage III vs. stage IV	0.182 (0.136–0.244)	<0.001	0.175 (0.13–0.235)	<0.001
Male vs. female sex	0.947 (0.763–1.177)	0.624	0.906 (0.726–1.131)	0.384
Right vs. left	1.119 (0.891–1.404)	0.332	1.116 (0.841–1.482)	0.447
Colon vs. rectum	0.963 (0.775–1.198)	0.736	0.937 (0.715–1.228)	0.637

**Table 4 ijms-25-08673-t004:** Interpretation of CDX2 immunohistochemical staining based on percentage of deeply stained nuclei.

Percentage of Deeply Stained Nuclei (%)	Score
Less than 5	1
5–24	2
25–49	3
50–74	4
75–100	5

## Data Availability

Data is available in the Appendix A.

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
