# Peer review of "The Prognostic and Predictive Utility of CDX2 in Colorectal Cancer"

_ijms, 2024, doi:10.3390/ijms25168673_

Round 1

Reviewer 1 Report

Comments and Suggestions for Authors

1.     In the introduction it is necessary to go into more detail about the molecular aspect of CDX2 as a biomarker to CRC, what happens when its expression is low or high, and how this can produce CRC

2.     In the discussion, the meaning of the low or high expression of CDX2 must also be discussed.

Author Response

Reviewer 1

  1. In the introduction it is necessary to go into more detail about the molecular aspect of CDX2 as a biomarker to CRC, what happens when its expression is low or high, and how this can produce CRC

  • We thank the reviewer for this insightful and very valid comment, and have included an entire section on the molecular characteristics of CDX2, its role in CRC, and its regulatory networks affecting carcinogenesis – lines 50 to 57, with 6 additional references.

  1. In the discussion, the meaning of the low or high expression of CDX2 must also be discussed.
  • Following on from point 1, we have also included the molecular mechanisms discussed earlier in the discussion – lines 231-233, 1 additional reference.

Reviewer 2 Report

Comments and Suggestions for Authors

The article approaches a highly important topic in patients with colorectal cancer and it suggests important implication of this biomarker in the patients' evolution.

Altough, the manuscript is generally well-written and designed, I suggest the authors to improve the discussions section since it is too short and it should involve more data from other studies reported in the literature on this topic.

Author Response

The article approaches a highly important topic in patients with colorectal cancer and it suggests important implication of this biomarker in the patients' evolution.

  • Thank you for this feedback.

Although, the manuscript is generally well-written and designed, I suggest the authors to improve the discussions section since it is too short and it should involve more data from other studies reported in the literature on this topic.

  • Thank you for this. To the best of our knowledge, we have included the available literature looking at the prognostic and predictive utility of CDX2 in CRC. However, we have added more information from other reviews that are contributory to our discussion – lines 256-257, 1 additional reference.  We have also incorporated additional information on molecular pathways in the background, into the discussion - lines 231-233, 1 additional reference; lines 354-356.